# Hindsight Credit Assignment

**Anna Harutyunyan, Will Dabney, Thomas Mesnard, Nicolas Heess, Mohammad G. Azar,
Bilal Piot, Hado van Hasselt, Satinder Singh, Greg Wayne, Doina Precup, Rémi Munos**
DeepMind
{harutyunyan, wdabney, munos}@google.com

## Abstract

We consider the problem of efficient credit assignment in reinforcement learning.
In order to efficiently and meaningfully utilize new data, we propose to explicitly
assign credit to past decisions based on the likelihood of them having led to the
observed outcome. This approach uses new information in hindsight, rather than
employing foresight. Somewhat surprisingly, we show that value functions can
be rewritten through this lens, yielding a new family of algorithms. We study the
properties of these algorithms, and empirically show that they successfully address
important credit assignment challenges, through a set of illustrative tasks.

## 1 Introduction

A reinforcement learning (RL) agent is tasked with two fundamental, interdependent problems:
exploration (how to discover useful data), and credit assignment (how to incorporate it). In this work,
we take a careful look at the problem of credit assignment. The instrumental learning object in RL –
the value function – quantifies the following question: *"how does choosing an action $a$ in a state $x$
affect future return?"*. This is a challenging question for several reasons.

**Issue 1: Variance.** The simplest way of estimating the value function is by averaging returns
(future discounted sums of rewards) starting from taking $a$ in $x$. This Monte Carlo style of estimation
is inefficient, since there can be a lot of randomness in trajectories.

**Issue 2: Partial observability.** To amortize the search and reduce variance, temporal difference
(TD) methods, like Sarsa and Q-learning, use a learned approximation of the value function and
bootstrap. This introduces bias due to the approximation, as well as a reliance on the Markov
assumption, which is especially problematic when the agent operates outside of a Markov Decision
Process (MDP), for example if the state is partially observed, or if there is function approximation.
Bootstrapping may then cause the value function to not converge at all, or to remain permanently
biased [19].

**Issue 3: Time as a proxy.** TD($\lambda$) methods control this bias-variance trade-off, but they rely on
*time* as the sole metric for relevance: the more recent the action, the more credit or blame it receives
from a future reward [20, 21]. Although time is a reasonable proxy for cause-and-effect (especially
in MDPs), in general it is a heuristic, and can hence be improved by learning.

**Issue 4: No counterfactuals.** The only data used for estimating an action's value are trajectories
that contain that action, while ideally we would like to be able to use the same trajectory to update *all*
relevant actions, not just the ones that happened to (serendipitously) occur.

Figure 1 illustrates these issues concretely. At the high-level, we wish to achieve credit assignment
mechanisms that are both sample-efficient (issues 1 and 4), and expressive (issues 2 and 3). To this
end, we propose to reverse the key learning question, and learn estimators that measure: *"given
the future outcome (reward or state), how relevant was the choice of $a$ in $x$ to achieve it?"*, which
is essentially the credit assignment question itself. Although eligibility traces consider the same

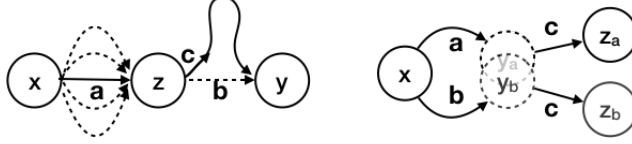

Figure 1: **Left.** Consider the trajectory shown by solid arrows to be the sampled trajectory, $\tau$. An RL algorithm will typically assign credit for the reward obtained in state $y$ to the actions along $\tau$. This is unsatisfying for two reasons: (1) action $a$ was not essential in reaching state $z$, any other $a'$ would have been just as effective; hence, overemphasizing $a$ is a source of variance; (2) from $z$, action $c$ was sampled, leading to a multi-step trajectory into $y$, but action $b$ transitions to $y$ from $z$ directly; so, it should get more of the credit for $y$. Note that $c$ could have been an exploratory action, but could also have been more likely according to the policy in $z$, but *given that $y$ was reached*, $b$ was more likely **Right.** The choice between actions $a$ or $b$ at state $x$ causes a transition to either $y_a$ or $y_b$, but they are perceptually aliased. On the next decision, the same action $c$ transitions the agent to different states, depending on the true underlying $y$. The state $y$ can be a single state, or could itself be a trajectory. This scenario can happen e.g. when the features are being learned. A TD algorithm that bootstraps in $y$ will not be able to learn the correct values of $a$ and $b$, since it will average over the rewards of $z_a$ and $z_b$. When $y$ is a potentially long trajectory with a noisy reward, a Monte Carlo algorithm will incorporate the noise along $y$ into the values of both $a$ and $b$, despite it being irrelevant to the choice between them. We would like to be able to directly determine the relevance of $a$ to being in $z_a$.

question, they do so in a way that is (purposefully) equivalent to the forward view [20], and so they have to rely mainly on "vanilla" features, like time, to decide credit assignment. Reasoning in the backward view explicitly opens up a new family of algorithms. Specifically, we propose to use a form of *hindsight conditioning* to determine the relevance of a past action to a particular outcome. We show that the usual value functions can be rewritten in hindsight, yielding a new family of estimators, and derive policy gradient algorithms that use these estimators. We demonstrate empirically the ability of these algorithms to address the highlighted issues through a set of diagnostic tasks, which are not handled well by other means.

## 2 Background and Notation

A Markov decision process (MDP) [14] is a tuple $(\mathcal{X}, \mathcal{A}, p, r, \gamma)$, with $\mathcal{X}$ being the state space, $\mathcal{A}$ - the action space, $p : \mathcal{X} \times \mathcal{A} \times \mathcal{X} \to [0, 1]$ – the state-transition distribution (with $p(y|x, a)$ denoting the probability of transitioning to state $y$ from $x$ by choosing action $a$), $r : \mathcal{X} \times \mathcal{A} \to \mathbb{R}$ – the reward function, and $\gamma \in [0, 1)$ – the scalar discount factor. A stochastic policy $\pi$ maps each state to a distribution over actions: $\pi(a|x)$ denotes the probability of choosing action $a$ in state $x$. Let $\mathcal{T}(x, \pi)$ and $\mathcal{T}(x, a, \pi)$ be the distributions over trajectories $\tau = (X_k, A_k, R_k)_{k \in \mathbb{N}^+}$ generated by a policy $\pi$, given $X_0 = x$ and $(X_0, A_0) = (x, a)$, respectively. Let $Z(\tau) \stackrel{\text{def}}{=} \sum_{k \geq 0} \gamma^k R_k$ be the return obtained along the trajectory $\tau$. The value (or V-) function $V^\pi$ and the action-value (or Q-) function $Q^\pi$ denote the expected return under the policy $\pi$ given $X_0 = x$ and $(X_0, A_0) = (x, a)$, respectively:

$$V^\pi(x) \stackrel{\text{def}}{=} \mathbb{E}_{\tau \sim \mathcal{T}(x,\pi)} \Big[ Z(\tau) \Big], \qquad Q^\pi(x, a) \stackrel{\text{def}}{=} \mathbb{E}_{\tau \sim \mathcal{T}(x,a,\pi)} \Big[ Z(\tau) \Big]. \tag{1}$$

The benefit of choosing a given action $a$ over the usual policy $\pi$ is measured by the advantage function $A^\pi(x, a) \stackrel{\text{def}}{=} Q^\pi(x, a) - V^\pi(x)$. Policy gradient algorithms improve the policy by changing $\pi$ in the direction of the gradient of the value function [22]. This gradient at some initial state $x_0$ is

$$\nabla V^\pi(x_0) = \sum_{x,a} d^\pi(x|x_0) Q^\pi(x, a) \nabla \pi(a|x) = \mathbb{E}_{\tau \sim \mathcal{T}(x_0,\pi)} \Big[ \sum_a \sum_{k \geq 0} \gamma^k A^\pi(X_k, a) \nabla \pi(a|X_k) \Big],$$

where $d^\pi(x|x_0) \stackrel{\text{def}}{=} \sum_k \gamma^k \mathbb{P}_{\tau \sim \mathcal{T}(x_0,\pi)}(X_k = x)$ is the (unnormalized) discounted state-visitation distribution. Practical algorithms such as REINFORCE [25] approximate $Q^\pi$ or $A^\pi$ with an $n$-step truncated return, possibly combined with a bootstrapped approximate value function $V$, which is also often used as baseline (see [22, 12]) along a trajectory $\tau = (X_k, A_k, R_k)_k \sim \mathcal{T}(x, a, \pi)$:

$$A^\pi(x, a) \approx \sum_{k=0}^{n-1} \gamma^k R_k + \gamma^n V(X_n) - V(x).$$

# 3 Conditioning on the Future

The classical value function attempts to answer the question: "how does the current action affect future outcomes?" By relying on predictions about these future outcomes, existing approaches often exacerbate problems around variance (issue 1) and partial observability (issue 2). Furthermore, these methods tend to use temporal distance as a proxy for relevance (issue 3) and are unable to assign credit counter-factually (issue 4). We propose to learn estimators that explicitly consider the credit assignment question: *"given an outcome, how relevant were past decisions?"*.

This approach can in fact be linked to some classical methods in statistical estimation. In particular, Monte Carlo simulation is known to be inaccurate when there are rare events that are of interest: the averaging requires an infeasible number of samples to obtain an accurate estimate [16]. One solution is to *change measures*, that is, to use another distribution for which the events are less rare, and correct with importance sampling. The Girsanov theorem is a well-known example of this in processes with Brownian dynamics [4], known to produce lower variance estimates.

This scenario of rare random events is particularly relevant to efficient credit assignment in RL. When a new significant outcome is experienced, the agent ought to quickly update its estimates and policy accordingly. Let $\tau \sim \mathcal{T}(x, \pi)$ be a sampled trajectory, and $f$ some function of it. By changing measures from the policy $\pi$ with which it was sampled to a future-conditional, or *hindsight* distribution $h(\cdot|x, \pi, f(\tau))$, we hope to improve the efficiency of credit assignment. The importance sampling ratio $\frac{h(a|x, \pi, f(\tau))}{\pi(a|x)}$ then precisely denotes the relevance of an action $a$ to the specific future $f(\tau)$. If the distribution $h(a|x, \pi, f(\tau))$ is accurate, this allows us to quickly assign credit to all actions relevant to achieving $f(\tau)$. In this work, we consider $f$ to be a future *state*, or a future *return*. To highlight the use of the future-conditional distribution, we refer to the resulting family of methods as Hindsight Credit Assignment (HCA).

The remainder of this section formalizes the insight outlined above, and derives the usual value functions and policy gradients in hindsight, while the next one presents new algorithms based on sampling these expressions.

## 3.1 Conditioning on Future States

The agent composes its estimates of the return from an action $a$ by summing over the rewards obtained from future states $X_k$. One option of hindsight conditioning is to consider, at each step, the likelihood of an action $a$ *given that the future state $X_k$ was reached.*

**Definition 1** (*State-conditional hindsight distributions*). *For any action $a$ and any state $y$, define $h_k(a|x, \pi, y)$ to be the conditional probability over trajectories $\tau \sim \mathcal{T}(x, \pi)$ of the first action $A_0$ of trajectory $\tau$ being equal to $a$, given that the state $y$ has occurred at step $k$ along trajectory $\tau$:*

$$h_k(a|x, \pi, y) \stackrel{def}{=} \mathbb{P}_{\tau \sim \mathcal{T}(x, \pi)}(A_0 = a|X_k = y). \tag{2}$$

Intuitively, $h_k(a|x, \pi, y)$ quantifies the relevance of action $a$ to the future state $X_k$. If $a$ is not relevant to reaching $X_k$, this probability is simply the policy $\pi(a|x)$ (there is no relevant information in $X_k$). If $a$ is instrumental to reaching $X_k$, $h_k(a|x, \pi, y) > \pi(a|x)$, and vice versa, if $a$ detracts from reaching $X_k$, $h_k(a|x, \pi, y) < \pi(a|x)$. In general, $h_k$ is a lower-entropy distribution than $\pi$. The relationship of $h_k$ to more familiar quantities can be understood through the following identity obtained by an application of Bayes' rule:

$$\frac{h_k(a|x, \pi, y)}{\pi(a|x)} = \frac{\mathbb{P}(X_k = y|X_0 = x, A_0 = a, \pi)}{\mathbb{P}(X_k = y|X_0 = x, \pi)} = \frac{\mathbb{P}_{\tau \sim \mathcal{T}(x, a, \pi)}(X_k = y)}{\mathbb{P}_{\tau \sim \mathcal{T}(x, \pi)}(X_k = y)}.$$

Using this identity and importance sampling, we can rewrite the usual Q-function in terms of $h_k$. Since there is only one policy $\pi$ involved here, we will drop the explicit conditioning, but it is implied.

**Theorem 1.** *Consider an action $a$ and a state $x$ for which $\pi(a|x) > 0$. Then the following holds:*

$$Q^\pi(x, a) = r(x, a) + \mathbb{E}_{\tau \sim \mathcal{T}(x, \pi)}\Big[\sum_{k \geq 1} \gamma^k \frac{h_k(a|x, X_k)}{\pi(a|x)} R_k\Big].$$

So, each of the rewards $R_k$ along the way is weighted by the ratio $\frac{h_k(a|x, X_k)}{\pi(a|x)}$, which exactly quantifies how relevant $a$ was in achieving the corresponding state $X_k$. Following the discussion above, this

ratio is 1 if $a$ is irrelevant, and larger or smaller than 1 in the other cases. The expression for the Q-function is similar to that in Eq. (1), but the new expectation is no longer conditioned on the initial action $a$ – the policy $\pi$ is followed from the start ($A_0 \sim \pi(\cdot|x)$ instead of $A_0 = a$). This is an important point, as it will allow us to use returns generated by *any* action $A_0$ to update the values of all actions, to the extent that they are relevant according to $\frac{h_k(a|x,X_k)}{\pi(a|x)}$. Theorem 1 implies the following expression for the advantage:

$$A^\pi(x,a) = r(x,a) - r^\pi(x) + \mathbb{E}_{\tau \sim \mathcal{T}(x,\pi)} \Big[ \sum_{k \geq 1} \Big( \frac{h_k(a|x,X_k)}{\pi(a|x)} - 1 \Big) \gamma^k R_k \Big], \qquad (3)$$

where $r^\pi(x) = \sum_{a \in \mathcal{A}} \pi(a|x) r(x,a)$. This form of the advantage is particularly appealing, since it directly removes irrelevant rewards from consideration. Indeed, whenever $\frac{h_k(a|x,X_k)}{\pi(a|x)} = 1$, the reward $R_k$ does not participate in the advantage for the value of action $a$. When there is inconsequential noise that is outside of the agent's control, this may greatly reduce the variance of the estimates.

**Removing time dependence.** For clarity of exposition, here we have considered the hindsight distribution to be additionally conditioned on time. Indeed, $h_k$ depends not only on reaching the state, but also on the number of timesteps $k$ that it takes to do so. In general, this can be limiting, as it introduces a stronger dependence on the particular trajectory, and a harder estimation problem of the hindsight distribution. It turns out we can generalize all of the results presented here to a *time-independent* distribution $h_\beta(a|x,y)$, which gives the probability of $a$ conditioned on reaching $y$ *at some point* in the future. The scalar $\beta \in [0,1)$ is the "probability of survival" at each step. This can either be the discount $\gamma$, or a termination probability if the problem is undiscounted. In the discounted reward case Eq. (3) can be written in terms of $h_\beta$ as follows:

$$A^\pi(x,a) = r(x,a) - r^\pi(x) + \mathbb{E}_{\tau \sim \mathcal{T}(x,\pi)} \Big[ \sum_{k \geq 1} \Big( \frac{h_\beta(a|x,X_k)}{\pi(a|x)} - 1 \Big) \gamma^k R_k \Big], \qquad (4)$$

with the choice of $\beta = \gamma$. The interested reader may find the relevant proofs in the appendix.

Finally, it is possible to obtain a hindsight V-function, analogously to the Q-function from Theorem 1. The next section does this for *return*-conditional HCA. We include other variations in appendix.

### 3.2 Conditioning on Future Returns

The previous section derived Q-functions that explicitly reweigh the rewards at each step, based on the corresponding states' connection to the action whose value we wish to estimate. Since ultimately we are interested in the return, we could alternatively use it for future conditioning itself.

**Definition 2** (*Return-conditional hindsight distributions*). *For any action $a$ and any possible return $z$, define $h_z(a|x,\pi,z)$ to be the conditional probability over trajectories $\tau \sim \mathcal{T}(x,\pi)$ of the first action $A_0$ being $a$, given that $z$ has been observed along $\tau$:*

$$h_z(a|x,\pi,z) \stackrel{def}{=} \mathbb{P}_{\tau \sim \mathcal{T}(x,\pi)}\big(A_0 = a | Z(\tau) = z\big).$$

The distribution $h_z(a|x,\pi,z)$ is intuitively similar to $h_k$, but instead of future states, it directly quantifies the relevance of $a$ to obtaining the entire return $z$. This is appealing, since in the end we care about returns. Further, this could be simpler to learn, since instead of the possibly high-dimensional state, we now need to worry only about a scalar outcome. On the other hand, it is no longer "jumpy" in time, so may benefit less from structure in the dynamics. As with $h_k$, we will drop the explicit conditioning on $\pi$, but it is implied. We have the following result.

**Theorem 2.** *Consider an action $a$, and assume that for any possible random return $z = Z(\tau)$ for some trajectory $\tau \sim \mathcal{T}(x,\pi)$ we have $h_z(a|x,z) > 0$. Then we have:*

$$V^\pi(x) = \mathbb{E}_{\tau \sim \mathcal{T}(x,a,\pi)} \Big[ Z(\tau) \frac{\pi(a|x)}{h_z(a|x,Z(\tau))} \Big]. \qquad (5)$$

The V- (rather than Q-) function form here has interesting properties that we will discuss in the next section. Mathematically, the two forms are analogous to derive, but the ratio is now flipped. Equations (5) and (1) imply the following expression for the advantage:

$$A^{\pi}(x, a) = \mathbb{E}_{\tau \sim \mathcal{T}(x,a,\pi)} \left[ \left( 1 - \frac{\pi(a|x)}{h_z(a|x, Z(\tau))} \right) Z(\tau) \right]. \tag{6}$$

The factor $c(a|x, Z) = 1 - \frac{\pi(a|x)}{h_z(a|x,Z)}$ expresses how much a single action $a$ contributed to obtaining a return $Z$. If other actions (drawn from $\pi(\cdot|x)$) would have yielded the same return, $c(a|x, Z) = 0$, and the advantage is 0. If an action $a$ has made achieving $Z$ more likely, then $c(a|x, Z) > 0$, and conversely, if other actions would have contributed to achieving $Z$ more than $a$, then $c(a|x, Z) < 0$. Hence, $c(a|x, Z)$ expresses the impact an action has on the environment, in terms of the return, if everything else (future decisions as well as randomness of the environment) is unchanged.

Both $h_\beta$ and $h_z$ can be learned online from sampled trajectories (see Sec. 4 for algorithms, and a discussion in Sec. 4.1). Finally, while we chose to focus on state and return conditioning, one could consider other options. For example, conditioning on the reward (instead of the state) at a future time $k$, or an embedding of (or part of) the future trajectory, could have interesting properties.

### 3.3 Policy Gradients

We now give a policy gradient theorem based on the new expressions of the value function.

**Theorem 3.** *Let $\pi_\theta$ be the policy parameterized by $\theta$, and $\beta = \gamma$. Then, the gradient of the value at some state $x_0$ is:*

$$\nabla_\theta V^{\pi_\theta}(x_0) = \mathbb{E}_{\tau \sim \mathcal{T}(x_0, \pi_\theta)} \left[ \sum_{k \geq 0} \gamma^k \sum_a \nabla \pi_\theta(a|X_k) Q^x(X_k, a) \right] \tag{7}$$

$$= \mathbb{E}_{\tau \sim \mathcal{T}(x_0, \pi_\theta)} \left[ \sum_{k \geq 0} \gamma^k \nabla \log \pi_\theta(A_k|X_k) A^z(X_k, A_k) \right], \tag{8}$$

$$Q^x(X_k, a) \stackrel{def}{=} r(X_k, a) + \sum_{t \geq k+1} \gamma^{t-k} \frac{h_\beta(a|X_k, X_t)}{\pi_\theta(a|X_k)} R_t,$$

$$A^z(x, a) \stackrel{def}{=} \left( 1 - \frac{\pi_\theta(a|x)}{h_z(a|x, Z(\tau_{k:\infty}))} \right) Z(\tau_{k:\infty}).$$

Note that the expression for state HCA in Eq. (7) is written for all actions, rather than only the sampled one. Interestingly, this form does not require (or benefit from) a baseline. Contrary to the usual all-actions algorithm which must use the critic, the HCA reweighting allows us to use returns sampled from a particular starting action to obtain value estimates for all actions.

## 4 Algorithms

Using the new policy gradient theorem 3, we will now give novel algorithms based on sampling the expectations (7) and (8). Then, we will discuss the training of the relevant hindsight distributions.

**State-Conditional HCA** Consider a parametric representation of the policy $\pi(\cdot|x)$ and the future-state-conditional distribution $h_\beta(a|x, y)$, as well as the baseline $V$ and an estimate of the immediate reward $\hat{r}$. Generate $T$-step trajectories $\tau^T = (X_s, A_s, R_s)_{0 \leq s \leq T}$. We can compose an estimate of the return for all actions $a$ (see Theorem 7 in appendix):

$$Q^x(X_s, a) \approx \hat{r}(X_s, a) + \sum_{t=s+1}^{T-1} \gamma^{t-s} \frac{h_\beta(a|X_s, X_t)}{\pi(a|X_s)} R_t + \gamma^{T-s} \frac{h_\beta(a|X_s, X_T)}{\pi(a|X_s)} V(X_T).$$

The algorithm proceeds by training $V(X_s)$ to predict the usual return $Z_s = \sum_{t=s}^{T-1} \gamma^{t-s} R_t + \gamma^{T-s} V(X_T)$ and $\hat{r}(X_s, A_s)$ to predict $R_s$ (square loss), the hindsight distribution $h_\beta(a|X_s, X_t)$ to predict $A_s$ (cross entropy loss), and finally by updating the policy logits with $\sum_a Q^x(X_s, a) \nabla \pi(a \mid X_s)$. See Algorithm 1 in appendix for the detailed pseudocode.

**Return-Conditional HCA** Consider a parametric representation of the policy $\pi(\cdot|x)$ and the return-conditioned distribution $h_z(a|x, z)$. Generate full trajectories $\tau = (X_s, A_s, R_s)_{s \in \mathbb{N}^+}$ and compute

the sampled advantage at each step:

$$A^z(X_s, A_s) = \left(1 - \frac{\pi(A_s|X_s)}{h_z(A_s|X_s, Z_s)}\right)Z_s,$$

where $Z_s = \sum_{t \geq s} \gamma^{t-s} R_t$. The algorithm proceeds by training the hindsight distribution $h_z(a|X_s, Z_s)$ to predict $A_s$ (cross entropy loss), and updating the policy gradient with $\nabla \log \pi(A_s \mid X_s) A^z(X_s, A_s)$. See Algorithm 2 in appendix for the detailed pseudocode.

**RL without value functions.** The return-conditional version lends itself to a particularly simple algorithm. In particular, we no longer need to learn the value function $V$ – if $h_z(a|X_s, Z_s)$ is estimated well, using complete rollouts is feasible without variance issues. This takes our idea of reversing the direction of the learning question to the extreme, it is now *entirely* in hindsight.

The result is an actor-critic algorithm, where the usual baseline $V(X_s)$ is replaced by $b_s \overset{\text{def}}{=} \frac{\pi(A_s|X_s)}{h_z(A_s|X_s, Z_s)} Z_s$. This baseline is strongly correlated to the return $Z_s$ (it is proportional to it), which is desirable since we would like to remove as much of the variance (due to the dynamics of the world, or the agent's own policy) as possible. The following proposition verifies that despite being correlated, this baseline does not introduce bias into the policy gradient.

**Proposition 1.** *The baseline $b_s = \frac{\pi(A_s|X_s)}{h_z(A_s|X_s, Z_s)} Z_s$ does not introduce any bias in the policy gradient:*

$$\mathbb{E}_{\tau \sim \mathcal{T}(x_0, \pi)}\left[\sum_s \gamma^s \nabla \log \pi(A_s|X_s)\big(Z_s(\tau) - b_s\big)\right] = \nabla V(x_0).$$

### 4.1 Learning Hindsight Distributions

We have given equivalent rewritings of the usual value functions in terms of the proposed hindsight distributions, and have motivated their properties, when they are accurate. Now, the question is if it is feasible to learn good estimates of those distributions from experience, and whether shifting the learning problem in this way is beneficial. The remainder of this section discusses this question, while the next one provides empirical evidence for the affirmative.

There are several conventional objects that could be learned to help with credit assignment: a value function, a forward model, or an inverse model over states. An accurate forward model allows one to compute value functions directly with no variance, and an accurate inverse model – to perform precise credit assignment. However, learning such generative models accurately is difficult and has been a long-standing challenge in RL, especially in high-dimensional state spaces. Interestingly, the hindsight distribution is a discriminative, rather than generative model, and is hence not required to model the full distribution over states. Additionally, the action space is usually much smaller than the state space, and so shifting the focus to actions potentially makes the problem much easier. When certain structure in the dynamics is present, learning hindsight distributions may be significantly easier still – e.g. if the transition model is stochastic or the policy is changing, a particular $(x, a)$ can lead to many possible future states, but a particular future state can be explained by a small number of past actions. In general, learning $h_z$ and $h_\beta$ are supervised learning problems, so the new algorithms delegate some of the learning difficulty in RL to a supervised setting, for which many efficient approaches exist (e.g. [7, 23]).

## 5 Experiments

To empirically validate our proposal in a controlled way, we devised a set of diagnostic tasks that highlight issues 1-4, while also being representative of what occurs in practice (Fig. 2). We then systematically verify the intuitions developed throughout the paper. In all cases, we learn the hindsight distributions in tandem with the control policy. For each problem we compare HCA with state and return conditioning to standard baseline policy gradient, that is: $n$-step advantage actor critic (with $n = \infty$ for Monte Carlo). All the results are an average of 100 independent runs, with the plots depicting means and standard deviations. For simplicity we take $\gamma = 1$ in all of the tasks.

**Shortcut.** We begin with an example capturing the intuition from Fig. 1 (left). Fig. 2 (left) depicts a chain of length $n$ with a rewarding final state. At each step, one action takes a shortcut and directly

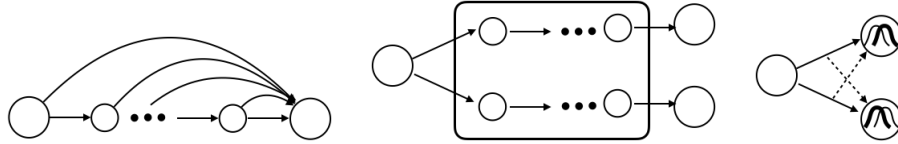

Figure 2: **Left:** Shortcut. Each state has two actions, one transitions directly to the goal, the other to the next state of the chain. **Center:** Delayed effect. Start state presents a choice of two actions, followed by an aliased chain, with the consequence of the initial choice apparent only in the final state. **Right:** Ambiguous bandit. Each action transitions to a particular state with high probability, but to the other action's state with low probability. When the two states have noisy rewards, credit assignment to each action becomes challenging.

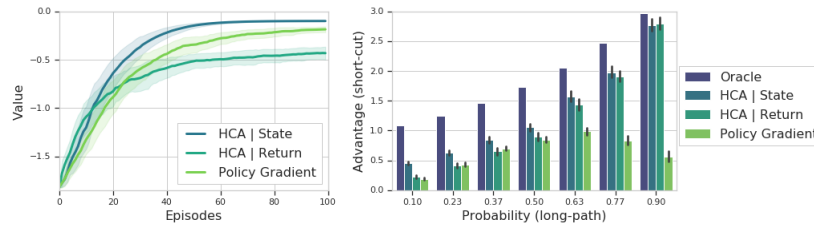

Figure 3: Shortcut. **Left:** learning curves for $n = 5$ with the policy between long and short paths initialized uniformly. Explicitly considering the likelihood of reaching the final state allows state-conditioned HCA to more quickly adjust its policy. **Right:** the advantage of the shortcut action estimated by performing 1000 rollouts from a fixed policy. The $x$-axis depicts the policy probabilities of the actions on the long path. The oracle is computed analytically without sampling. When the shortcut action is unlikely and rarely encountered, it is difficult to obtain an accurate estimate of the advantage. HCA is consistently able to maintain larger (and more accurate) advantages.

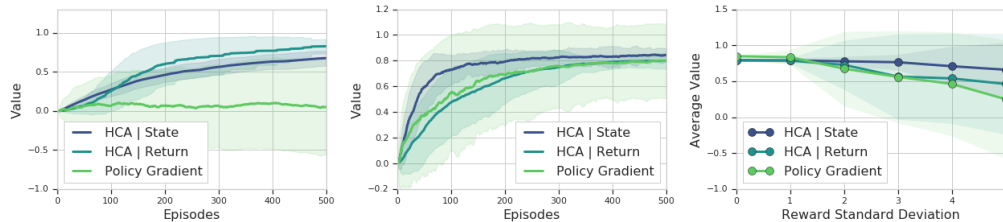

Figure 4: Delayed effect. **Left:** Bootstrapping. The learning curves for $n = 5$, $\sigma = 0$, and a 3-step return, which causes the agent to bootstrap in the partially observed region. As expected, naive bootstrapping is unable to learn a good estimate. **Middle:** Using full Monte Carlo returns (for $n = 3$) overcomes partial observability, but is prone to noise. The plot depicts learning curves for the setting with added white noise of $\sigma = 2$. **Right.** The average performance w.r.t. different noise levels – predictably, state HCA is the most robust.

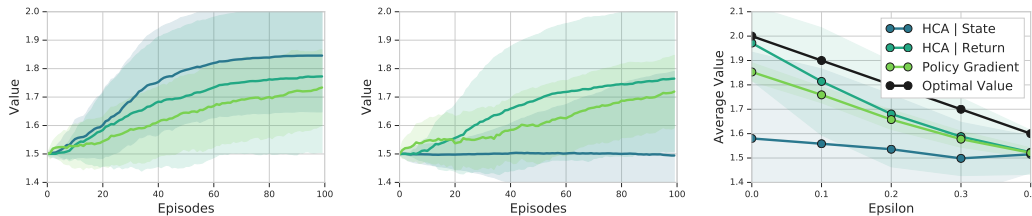

Figure 5: Ambiguous bandit with Gaussian rewards of means 1, 2, and standard deviation 1.5. **Left:** The state identity is observed. Both HCA methods improve on PG. **Middle:** The state identity is hidden, handicapping state HCA, but return HCA continues to improve on PG. **Right:** Average performance w.r.t. different $\epsilon$-s with Gaussian rewards of means 1, 2, and standard deviation 0.5. Note that the optimal value itself decays in this case.

transitions to the final state, while the other continues on the longer path, which may be more likely according to the policy. There is a per-step penalty (of $-1$), and a final reward of $1$. There is also a chance (of $0.1$) that the agent transitions to the absorbing state directly.

This problem highlights two issues: (1) the importance of counter-factual credit assignment (issue 4); when the long path is taken more frequently than the shortcut path, counter-factual updates become increasingly effective (see Fig. 3, right) (2) the use of time as a proxy for relevance (issue 3) is shown to be only a heuristic, even in a fully-observable MDP. The relevance for the states along the chain is not accurately reflected in the long temporal distance between them and the goal state. In Fig. 3 we show that HCA is more effective at quickly adjusting the policy towards the shortcut action.

**Delayed Effect.**  The next task instantiates the example from Fig. 1 (right). Fig. 2 (middle) depicts a POMDP, in which after the first decision, there is aliasing until the final state. This is a common case of partial observability, and is especially pertinent if the features are being learned. We show that (1) Bootstrapping naively is inadequate in this case (issue 2), but HCA is able to carry the appropriate information;[1] and (2) While Monte Carlo is able to overcome the partial observability, its performance deteriorates when intermediate reward noise is present (issue 1). HCA on the other hand is able to reduce the variance due to the irrelevant noise in the rewards.

Additionally, in this example the first decision is the most relevant choice, despite being the most temporally remote, once again highlighting that using temporal proximity for credit assignment is a heuristic (issue 3). One of the final states is rewarding (with $r = 1$), the other penalizing (with $r = -1$), and the middle states contain white noise of standard deviation $\sigma$. Fig. 4 depicts our results. In this task, the return-conditional HCA has a more difficult learning problem, as it needs to correctly model the noise distribution to condition on, which is as difficult as learning the values naively, and hence performs similarly to the baseline.

**Ambiguous Bandit.**  Finally, to emphasize that credit assignment can be challenging, even when it is not long-term, we consider a problem without a temporal component. Fig. 2 (right) depicts a bandit with two actions, leading to two different states, whose reward functions are similar (here: drawn from overlapping Gaussian distributions), with some probability $\epsilon$ of crossover. The challenge here is due to variance (issue 1) and a lack of counter-factual updates (issue 4). It is difficult to tell whether an action was genuinely better, or just happened to be on the tail end of the distribution. This is a common scenario when bootstrapping with similar values. Due to the explicit aim at modeling the distributions, the hindsight algorithms are more efficient (Fig. 5 (left)).

To highlight the differences between the two types of hindsight conditioning, we introduce partial observability (issue 2), see Fig. 5 (right). The return-conditional policy is still able to improve over policy gradient, but state-conditioning now fails to provide informative conditioning (by construction).

## 6   Related Work

Hindsight experience replay (HER) [1] introduces the idea of off-policy learning about many goals from the same trajectory. The intuition is that regardless of what goal the trajectory was pursuing originally, *in hindsight* it, e.g., successfully found the one corresponding to its final state, and there is something to be learned. Rauber et al. [15] extend the same intuition to policy gradient algorithms, with goal-conditioned policies. Goyal et al. [5] also use goal conditioning and learn a backtracking model, which predicts the state-action pairs occurring on trajectories that end up in goal states. These works share our intuition of in hindsight using the same data to learn about many things, but in the context of goal-conditioned policies, while we essentially contrast conditional and unconditional policies, where the conditioning is on the extra outcome (state or return). Note that we never act w.r.t. the conditional policy, and it is used solely for credit assignment.

The temporal value transport algorithm [11] also aims to propagate credit efficiently backward in time. It uses an attention mechanism over memory to jump over parts of a trajectory that are irrelevant for the rewards obtained. While demonstrated on challenging problems, that method is biased; a promising direction for future research is to apply our unbiased hindsight mechanism with past states chosen by such an attention mechanism. Another line of work with a related intuition is RUDDER [2]. It uses an LSTM to predict future returns and sensitivity analysis to distribute those

returns as immediate rewards in order to reduce the learning horizon and make long-term credit assignment easier. Instead of aiming to redistribute the return, state HCA up- or dowmnweights individual rewards according to their relevance to the past action.

A large number of variance reduction techniques have been applied in RL, e.g. using learned value functions as critics, and other control variates [e.g. 24]. When a model of the environment is available, it can be used to reduce variance. Rollouts from the same state fill the same role in policy gradients [18]. Differentiable system dynamics allow low-variance estimates of the Q-value gradient by using the pathwise derivative estimator, effectively backpropagating the gradient of the objective along trajectories [e.g. 17, 9, 10]. In stochastic systems this requires knowledge of the environment noise. To bypass this, Heess et al. [9] *infer* the noise given an observed trajectory. Buesing et al. [3] apply this idea to POMDPs, where it can be viewed as reasoning about events in hindsight. They use a structural causal model of the dynamics and infer the posterior over latent causes from empirical trajectories. Using an empirical rather than a learned distribution over latent causes can reduce bias and, together with the (deterministic) model of the system dynamics, allows exploring the effect of alternative action choices for an observed trajectory.

Inverse models similar to the ones we use appear, for instance, in variational intrinsic control [6] (see also e.g. [8]). However, in our work, the inverse model serves as a way of determining the influence of an action on a future outcome, whereas the work in [6, 8] aims to use the inverse model to derive an intrinsic reward for training policies in which actions influence the future observations.

Finally, prioritized sweeping can be viewed as changing the sampling distribution with hindsight knowledge of the TD errors [13].

## 7 Closing

We proposed a new family of algorithms that explicitly consider the question of credit assignment as a part of, or instead of, estimating the traditional value function. The proposed estimators come with new properties, and as we validate empirically, are able to address some of the key issues in credit assignment. Investigating the scalability of these algorithms in the deep reinforcement learning setting is an exciting problem for future research.

**Acknowledgements**

The authors thank Joseph Modayil for reviews of earlier manuscripts, Theo Weber for several insightful suggestions, and the anonymous reviewers for their useful feedback.

## Footnotes

[1]See the discussion in Appendix F.

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
