[Supplementary Material]

# A Proofs

## A.1 Proof of Theorem 1

**Lemma 1.** *For any initial state $x$, a state $y$ that can occur on a trajectory $\tau \sim \mathcal{T}(x, \pi)$, that is: $\mathbb{P}_{\tau \sim \mathcal{T}(x,\pi)}(X_k = y) \neq 0$ for some $k$ an action $a$ for which $\pi(a|x) \neq 0$, we have:*

$$\frac{h_k(a|x,y)}{\pi(a|x)} = \frac{\mathbb{P}_{\tau \sim \mathcal{T}(x,\pi)}(X_k = y|A_0 = a)}{\mathbb{P}_{\tau \sim \mathcal{T}(x,\pi)}(X_k = y)}. \tag{9}$$

*Proof.* From Bayes' rule, we have:

$$\mathbb{P}_{\tau \sim \mathcal{T}(x,\pi)}(X_k = y|A_0 = a) = \frac{\mathbb{P}_{\tau \sim \mathcal{T}(x,\pi)}(A_0 = a|X_k = y)\mathbb{P}_{\tau \sim \mathcal{T}(x,\pi)}(X_k = y)}{\mathbb{P}_{\tau \sim \mathcal{T}(x,\pi)}(A_0 = a)},$$

$$= \frac{\mathbb{P}_{\tau \sim \mathcal{T}(x,\pi)}(X_k = y)h_k(a|x,y)}{\pi(a|x)}.$$

$\square$

*Proof of Theorem 1.* From the definition of the Q-function for a state-action pair $(x, a)$, we have

$$Q^\pi(x,a) = r(x,a) + \sum_{k \geq 1}\sum_{y \in \mathcal{X}} \gamma^k \mathbb{P}_{\tau \sim \mathcal{T}(x,\pi)}(X_k = y|A_0 = a)r^\pi(y), \tag{10}$$

where $r^\pi(y) = \sum_{a \in \mathcal{A}} \pi(a|y)r(y,a)$.

Combining Eq. (9) with Eq. (10) we deduce

$$Q^\pi(x,a) = r(x,a) + \sum_{y \in \mathcal{X}}\sum_{k \geq 1} \gamma^k \mathbb{P}_{\tau \sim \mathcal{T}(x,\pi)}(X_k = y)\frac{h_k(a|x,y)}{\pi(a|x)}r^\pi(y),$$

$$= r(x,a) + \mathbb{E}_{\tau \sim \mathcal{T}(x,\pi)}\left[\sum_{k \geq 1}\gamma^k \frac{h_k(a|X_k,x)}{\pi(a|x)}R_k\right].$$

$\square$

## A.2 Proof of Theorem 2

*Proof.* For any action $a$, the value function writes as

$$
\begin{aligned}
V^\pi(x) &= \mathbb{E}_{\tau \sim \mathcal{T}(x,\pi)}\big[Z(\tau)\big], \\
&= \int_z z\mathbb{P}_{\tau \sim \mathcal{T}(x,\pi)}(Z(\tau) = z)dz, \\
&= \int_z z\frac{\mathbb{P}_{\tau \sim \mathcal{T}(x,\pi)}(Z(\tau) = z)}{\mathbb{P}_{\tau \sim \mathcal{T}(x,a,\pi)}(Z(\tau) = z)}\mathbb{P}_{\tau \sim \mathcal{T}(x,a,\pi)}(Z(\tau) = z)dz, \\
&= \int_z z\frac{\mathbb{P}_{\tau \sim \mathcal{T}(x,\pi)}(Z(\tau) = z)}{\mathbb{P}_{\tau \sim \mathcal{T}(x,\pi)}(Z(\tau) = z|A_0 = a)}\mathbb{P}_{\tau \sim \mathcal{T}(x,a,\pi)}(Z(\tau) = z)dz, \\
&\stackrel{(i)}{=} \int_z z\frac{\mathbb{P}_{\tau \sim \mathcal{T}(x,\pi)}(A_0 = a)}{\mathbb{P}_{\tau \sim \mathcal{T}(x,\pi)}(A_0 = a|Z(\tau) = z)}\mathbb{P}_{\tau \sim \mathcal{T}(x,a,\pi)}(Z(\tau) = z)dz, \\
&= \int_z z\frac{\pi(a|x)}{h_z(a|x,z)}\mathbb{P}_{\tau \sim \mathcal{T}(x,a,\pi)}(Z(\tau) = z)dz, \\
&= \mathbb{E}_{\tau \sim \mathcal{T}(x,a,\pi)}\left[Z(\tau)\frac{\pi(a|x)}{h_z(a|x,Z(\tau))}\right],
\end{aligned}
$$

where $(i)$ follows from Bayes' rule.

$\square$

### A.3 Proof of Theorem 3

*Proof.* Using (3), we have:

$$\nabla_\theta V^{\pi_\theta}(x_0) = \mathbb{E}_{\tau \sim \mathcal{T}(x_0,\pi_\theta)}\Big[\sum_a \sum_{k\geq 0} \gamma^k \nabla \pi_\theta(a|X_k) A^\pi(X_k, a)\Big]$$

$$= \mathbb{E}_{\tau \sim \mathcal{T}(x_0,\pi_\theta)}\Big[\sum_a \sum_{k\geq 0} \gamma^k \nabla \pi_\theta(a|X_k)\Big(r(X_k,a) - r^{\pi_\theta}(X_k) + \sum_{t\geq k+1} \gamma^{t-k}\Big(\frac{h_\beta(a|X_k, X_t)}{\pi_\theta(a|X_k)} - 1\Big)R_t\Big)\Big]$$

$$= \mathbb{E}_{\tau \sim \mathcal{T}(x_0,\pi_\theta)}\Big[\sum_a \sum_{k\geq 0} \gamma^k \nabla \pi_\theta(a|X_k)\Big(r(X_k,a) + \sum_{t\geq k+1} \gamma^{t-k}\frac{h_\beta(a|X_k, X_t)}{\pi_\theta(a|X_k)}R_t\Big)\Big].$$

where the third equality is due to $\sum_a \nabla \pi_\theta(a|X_k) f(X_k) = f(X_k) \sum_a \nabla \pi_\theta(a|X_k) = 0$, for $f(X_k) = r^{\pi_\theta}(X_k) + \sum_{t\geq k+1} \gamma^{t-k} R_t$.

Similarly, for the return version and any action $a$, we have:

$$\nabla_\theta V^{\pi_\theta}(x_0) = \mathbb{E}_{\tau \sim \mathcal{T}(x_0,\pi_\theta)}\Big[\sum_a \sum_{k\geq 0} \gamma^k \nabla \pi_\theta(a|X_k) A^\pi(X_k, a)\Big]$$

$$= \mathbb{E}_{\tau \sim \mathcal{T}(x_0,\pi_\theta)}\Big[\sum_a \sum_{k\geq 0} \gamma^k \pi(a|X_k) \nabla \log \pi_\theta(a|X_k) A^\pi(X_k, a)\Big]$$

$$= \mathbb{E}_{\tau \sim \mathcal{T}(x_0,\pi_\theta)}\Big[\sum_{k\geq 0} \gamma^k \nabla \log \pi_\theta(A_k|X_k) A^\pi(X_k, A_k)\Big]$$

$$= \mathbb{E}_{\tau \sim \mathcal{T}(x_0,\pi_\theta)}\Big[\sum_{k\geq 0} \gamma^k \nabla \log \pi_\theta(A_k|X_k)\Big(1 - \frac{\pi(A_k|X_k)}{h_z(A_k|X_k, Z(\tau_{k:\infty}))}\Big)Z(\tau_{k:\infty})\Big].$$

$\square$

### A.4 Proof of Proposition 1

*Proof.* We have:

$$\mathbb{E}_{\tau \sim \mathcal{T}(x_0,\pi)}\Big[\sum_s \gamma^s \nabla \log \pi(A_s|X_s)\big(Z_s(\tau) - b_s\big)\Big]$$

$$= \mathbb{E}_{\tau \sim \mathcal{T}(x_0,\pi)}\Big[\sum_s \gamma^s \nabla \log \pi(A_s|X_s) Q^\pi(X_s, A_s)\Big] - \mathbb{E}_{\tau \sim \mathcal{T}(x_0,\pi)}\Big[\nabla \log \pi(A_s|X_s) b_s\Big],$$

$$= \nabla V(x_0) - \mathbb{E}_{\tau \sim \mathcal{T}(x_0,\pi)}\Big[\nabla \log \pi(A_s|X_s) \frac{\pi(A_s|X_s)}{h_z(A_s|X_s, Z_s(\tau))} Z_s(\tau)\Big],$$

$$\overset{(i)}{=} \nabla V(x_0) - \mathbb{E}_{\tau \sim \mathcal{T}(x_0,\pi)}\Big[\mathbb{E}_{A_s \sim \pi(\cdot|X_s)}\Big[\nabla \log \pi(A_s|X_s) \underbrace{\mathbb{E}_{\tau \sim \mathcal{T}(X_s, A_s, \pi)}\Big[\frac{\pi(A_s|X_s)}{h_z(A_s|X_s, Z_s(\tau))} Z_s(\tau)\Big]}_{V^\pi(X_s)}\Big]\Big],$$

$$= \nabla V(x_0) - \mathbb{E}_{\tau \sim \mathcal{T}(x_0,\pi)}\Big[V^\pi(X_s) \sum_{a\in\mathcal{A}} \nabla \pi(a|X_s)\Big],$$

$$= \nabla V(x_0).$$

where $(i)$ follows from Theorem 2. $\square$

## B  Other variants

Analogously to Theorems 1 and 2, we can obtain the V- and Q-functions for state and return conditioning, respectively. We have:

**Theorem 4.** *Consider an action $a$ for which $\pi(a|x) > 0$ and $\mathbb{P}_{\tau \sim \mathcal{T}(x,\pi)}(X_k = y|A_0 = a) > 0$ for any state $X_k$ sampled on $\tau \sim \mathcal{T}(x, a, \pi)$:*

$$V^\pi(x) = \mathbb{E}_{\tau \sim \mathcal{T}(x,a,\pi)}\Big[\sum_{k\geq 0} \gamma^k \frac{\pi(a|x)}{h_k(a|x, X_k)} R_k\Big].$$

*Proof.* We can flip the result of Lemma 1 for actions $a$ for which $\pi(a|x) > 0$ and $\mathbb{P}_{\tau \sim \mathcal{T}(x,\pi)}(X_k = y|A_0 = a) > 0$.

$$\frac{\pi(a|x)}{h_k(a|x,y)} = \frac{\mathbb{P}_{\tau \sim \mathcal{T}(x,\pi)}(X_k = y)}{\mathbb{P}_{\tau \sim \mathcal{T}(x,\pi)}(X_k = y|A_0 = a)}. \tag{11}$$

Let $r^\pi(y) = \sum_{a \in \mathcal{A}} \pi(a|y) r(y,a)$. We have

$$
\begin{aligned}
V^\pi(x) &= \mathbb{E}_{\tau \sim \mathcal{T}(x,\pi)}\Big[\sum_{k \geq 0} \gamma^k R_k\Big] \\
&= \sum_{k \geq 0}\sum_{y \in \mathcal{X}} \gamma^k \mathbb{P}_{\tau \sim \mathcal{T}(x,\pi)}(X_k = y) r^\pi(y) \\
&= \sum_{k \geq 0}\sum_{y \in \mathcal{X}} \gamma^k \mathbb{P}_{\tau \sim \mathcal{T}(x,\pi)}(X_k = y|A_0 = a)\frac{\mathbb{P}_{\tau \sim \mathcal{T}(x,\pi)}(X_k = y)}{\mathbb{P}_{\tau \sim \mathcal{T}(x,\pi)}(X_k = y|A_0 = a)} r^\pi(y) \\
&= \sum_{k \geq 0}\sum_{y \in \mathcal{X}} \gamma^k \mathbb{P}_{\tau \sim \mathcal{T}(x,\pi)}(X_k = y|A_0 = a)\frac{\pi(a|x)}{h_k(a|x,y)} r^\pi(y) \\
&= \mathbb{E}_{\tau \sim \mathcal{T}(x,a,\pi)}\Big[\sum_{k \geq 0} \gamma^k \frac{\pi(a|x)}{h_k(a|x,X_k)} R_k\Big].
\end{aligned}
$$

$\square$

**Theorem 5.** *Consider an action $a$ for which $\pi(a|x) > 0$. We have:*

$$Q^\pi(x,a) = \mathbb{E}_{\tau \sim \mathcal{T}(x,\pi)}\Big[Z(\tau)\frac{h_z(a|x,Z(\tau))}{\pi(a|x)}\Big]. \tag{12}$$

*Proof.* The Q-function writes:

$$
\begin{aligned}
Q^\pi(x,a) &= \mathbb{E}_{\tau \sim \mathcal{T}(x,a,\pi)}\big[Z(\tau)\big], \\
&= \int_z z\,\mathbb{P}_{\tau \sim \mathcal{T}(x,a,\pi)}(Z(\tau) = z)dz, \\
&= \int_z z\frac{\mathbb{P}_{\tau \sim \mathcal{T}(x,a,\pi)}(Z(\tau) = z)}{\mathbb{P}_{\tau \sim \mathcal{T}(x,\pi)}(Z(\tau) = z)}\mathbb{P}_{\tau \sim \mathcal{T}(x,\pi)}(Z(\tau) = z)dz, \\
&= \int_z z\frac{\mathbb{P}_{\tau \sim \mathcal{T}(x,\pi)}(Z(\tau) = z|A_0 = a)}{\mathbb{P}_{\tau \sim \mathcal{T}(x,\pi)}(Z(\tau) = z)}\mathbb{P}_{\tau \sim \mathcal{T}(x,\pi)}(Z(\tau) = z)dz, \\
&\overset{(i)}{=} \int_z z\frac{\mathbb{P}_{\tau \sim \mathcal{T}(x,\pi)}(A_0 = a|Z(\tau) = z)}{\mathbb{P}_{\tau \sim \mathcal{T}(x,\pi)}(A_0 = a)}\mathbb{P}_{\tau \sim \mathcal{T}(x,\pi)}(Z(\tau) = z)dz, \\
&= \int_z z\frac{h_z(a|x,z)}{\pi(a|x)}\mathbb{P}_{\tau \sim \mathcal{T}(x,\pi)}(Z(\tau) = z)dz, \\
&= \mathbb{E}_{\tau \sim \mathcal{T}(x,\pi)}\Big[Z(\tau)\frac{h_z(a|x,Z(\tau))}{\pi(a|x)}\Big],
\end{aligned}
$$

where $(i)$ follows from Bayes' rule. $\square$

## C  Time-Independent State-Conditional Case

We begin by introducing a time independent variant of state-conditional distribution. Let $\beta \in [0,1)$ and $\rho(k) = \beta^{k-1}(1 - \beta)$ be the geometric distribution on $k \in \mathbb{N}^+$. Then the state-conditional distribution $h_\beta(a|y,x)$ writes as follows for a future state $y$:

$$h_\beta(a|x,y) \overset{\text{def}}{=} \mathbb{P}_{\tau \sim \mathcal{T}(x,\pi)}(A_0 = a|X_k = y, k \sim \rho). \tag{13}$$

We draw the attention of readers to the difference between the new definition of $h_\beta$ and the original one in Eq. 2: in this case the timestep $k$ is a random event drawn from the distribution $\rho$, whereas in Eq. 2 the timestep $k$ is a fixed scalar.

We now show that the result of Theorem 1 extends to the case of $h_\beta$ with the choice of $\beta = \gamma$.

**Theorem 6.** *Consider an action $a$ and a state $x$ for which $\pi(a|x)>0$. Set the scalar $\beta = \gamma$. Then $Q^\pi$ writes as*

$$Q^\pi(x,a) = r(x,a) + \mathbb{E}_{\tau \sim \mathcal{T}(x,\pi)}\Big[\sum_{k \geq 1} \gamma^k \frac{h_\beta(a|x, X_k)}{\pi(a|x)} R_k\Big].$$

*Proof.* Let us introduce the coefficient $c_\gamma = \frac{\gamma}{1-\gamma}$ such that $c_\gamma \rho(k) = \gamma^k$. By definition of the Q-function for a state-action couple $(x,a)$, we have

$$Q^\pi(x,a) = r(x,a) + \sum_{k \geq 1}\sum_{y \in \mathcal{X}} \gamma^k \mathbb{P}_{\tau \sim \mathcal{T}(x,\pi)}(X_k = y|A_0 = a)r^\pi(y),$$

which can be rewritten:

$$Q^\pi(x,a) = r(x,a) + c_\gamma \sum_{y \in \mathcal{X}}\sum_{k \geq 1} \rho(k)\mathbb{P}_{\tau \sim \mathcal{T}(x,\pi)}(X_k = y|A_0 = a)r^\pi(y). \tag{14}$$

From the law of total probability and the independence between the events $k \sim \rho$ and $A_0 = a$:

$$\mathbb{P}_{\tau \sim \mathcal{T}(x,\pi)}(X_k = y|A_0 = a, k \sim \rho) = \sum_{k \geq 1} \rho(k)\mathbb{P}_{\tau \sim \mathcal{T}(x,\pi)}(X_k = y|A_0 = a).$$

Combining this with Eq. (14) we deduce

$$Q^\pi(x,a) = r(x,a) + c_\gamma \sum_{y \in \mathcal{X}} \mathbb{P}_{\tau \sim \mathcal{T}(x,\pi)}(X_k = y|A_0 = a, k \sim \rho)r^\pi(y). \tag{15}$$

From applying the Bayes' rule and independence between the events $k \sim \rho$ and $A_0 = a$, we have

$$\mathbb{P}_{\tau \sim \mathcal{T}(x,\pi)}(X_k = y|A_0 = a, k \sim \rho) = \frac{h_\beta(a|x,y)\mathbb{P}_{\tau \sim \mathcal{T}(x,\pi)}(X_k = y|k \sim \rho)}{\pi(a|x)}.$$

Combining this with Eq. (15) we deduce

$$Q^\pi(x,a) = r(x,a) + c_\gamma \sum_{y \in \mathcal{X}} \mathbb{P}_{\tau \sim \mathcal{T}(x,\pi)}(X_k = y|k \sim \rho)\frac{h_\beta(a|x,y)}{\pi(a|x)}r^\pi(y),$$

$$= r(x,a) + \sum_{y \in \mathcal{X}}\sum_{k \geq 1} \gamma^k \mathbb{P}_{\tau \sim \mathcal{T}(x,\pi)}(X_k = y)\frac{h_\beta(a|x,y)}{\pi(a|x)}r^\pi(y),$$

$$= r(x,a) + \mathbb{E}_{\tau \sim \mathcal{T}(x,\pi)}\Big[\sum_{k \geq 1}\gamma^k \frac{h_\beta(a|X_k, x)}{\pi(a|x)}r^\pi(X_k)\Big],$$

$$= r(x,a) + \mathbb{E}_{\tau \sim \mathcal{T}(x,\pi)}\Big[\sum_{k \geq 1}\gamma^k \frac{h_\beta(a|X_k, x)}{\pi(a|x)}R_k\Big].$$

$\square$

We now extend the result of Theorem 6 to the case of $T$-step bootstrapped return. Let $\rho_T$ be the distribution on the set $\{1, 2, \ldots, T\}$ defined as

$$\rho_T(k) \stackrel{\text{def}}{=} \begin{cases} \beta^{k-1}(1-\beta) & 1 \leq k < T \\ \beta^{T-1} & k = T \end{cases} \tag{16}$$

We also define the $T$-step state-conditional distribution $h_{\beta,T}(a|y, x)$ for a future state $y$:

$$h_{\beta,T}(a|x,y) \stackrel{\text{def}}{=} \mathbb{P}_{\tau \sim \mathcal{T}(x,\pi)}(A_0 = a|X_k = y, k \sim \rho_T). \tag{17}$$

**Theorem 7.** *Consider an action $a$ and a state $x$ for which $\pi(a|x) > 0$. Set the scalar $\beta = \gamma$. Then $Q^\pi$ writes as*

$$Q^\pi(x,a) = r(x,a) + \mathbb{E}_{\tau \sim \mathcal{T}(x,\pi)} \Big[ \sum_{k \geq 1}^{T-1} \gamma^k \frac{h_{\beta,T}(a|x,X_k)}{\pi(a|x)} R_k + \gamma^T \frac{h_{\beta,T}(a|x,X_T)}{\pi(a|x)} V^\pi(X_T) \Big].$$

*Proof.* By definition of the Q-function for a state-action couple $(x,a)$, we have

$$Q^\pi(x,a) = r(x,a) + \sum_{k=1}^{T-1} \sum_{y \in \mathcal{X}} \gamma^k \mathbb{P}_{\tau \sim \mathcal{T}(x,\pi)}(X_k = y | A_0 = a) r^\pi(y) + \sum_{y \in \mathcal{X}} \gamma^T \mathbb{P}_{\tau \sim \mathcal{T}(x,\pi)}(X_T = y | A_0 = a) V^\pi(y),$$

From the definition of the (normalized) discounted visit distribution $\tilde{d}^\pi(z|y) \stackrel{\text{def}}{=} (1 - \gamma) \sum_k \gamma^k \mathbb{P}_{\tau \sim \mathcal{T}(y,\pi)}(X_k = z)$, we have:

$$V^\pi(y) = \frac{1}{1-\gamma} \sum_{z \in \mathcal{X}} \tilde{d}^\pi(z|y) r^\pi(z).$$

Therefore $Q^\pi(x,a)$ can be rewritten:

$$Q^\pi(x,a) = r(x,a) + \sum_{k=1}^{T-1} \sum_{y \in \mathcal{X}} \gamma^k \mathbb{P}_{\tau \sim \mathcal{T}(x,\pi)}(X_k = y | A_0 = a) r^\pi(y)$$

$$+ \frac{\gamma^T}{1-\gamma} \sum_{y \in \mathcal{X}} \sum_{z \in \mathcal{X}} \mathbb{P}_{\tau \sim \mathcal{T}(x,\pi)}(X_T = y | A_0 = a) \tilde{d}^\pi(z|y) r^\pi(z).$$

Now let us define the following distribution $\mu_k(.|y)$ for each $(k,y)$:

$$\mu_k(z|y) \stackrel{\text{def}}{=} \begin{cases} \mathbf{1}_{z=y} & 1 \leq k < T \\ \tilde{d}^\pi(z|y) & k = T. \end{cases} \tag{18}$$

Thus we can rewrite $Q^\pi(x,a)$ as:

$$Q^\pi(x,a) = r(x,a) + c_\gamma \sum_{k=1}^{T} \sum_{y \in \mathcal{X}} \sum_{z \in \mathcal{X}} \rho_T(k) \mathbb{P}_{\tau \sim \mathcal{T}(x,\pi)}(X_k = y | A_0 = a) \mu_k(z|y) r^\pi(z).$$

From the law of total probability, independence between the events $k \sim \rho_T$ and $A_0 = a$ and the Markovian relation between $X_k$ and $Z_k$ ($Z_k$ is a random variable with distribution $\mu_k(.|X_k)$):

$$\mathbb{P}_{\tau \sim \mathcal{T}(x,\pi)}(X_k = y, Z_k = z | A_0 = a, k \sim \rho_T) = \sum_{k=1}^{T} \rho_T(k) \mathbb{P}_{\tau \sim \mathcal{T}(x,\pi)}(X_k = y, Z_k = z | A_0 = a),$$

$$= \sum_{k \geq 1} \rho_T(k) \mathbb{P}_{\tau \sim \mathcal{T}(x,\pi)}(X_k = y | A_0 = a) \mu_k(Z_k = z | X_k = y).$$

Therefore we have:

$$Q^\pi(x,a) = r(x,a) + c_\gamma \sum_{y \in \mathcal{X}} \sum_{z \in \mathcal{X}} \mathbb{P}_{\tau \sim \mathcal{T}(x,\pi)}(X_k = y, Z_k = z | A_0 = a, k \sim \rho_T) r^\pi(z).$$

Then, by applying the Bayes' rule:

$$\frac{\mathbb{P}_{\tau \sim \mathcal{T}(x,\pi)}(X_k = y, Z_k = z | A_0 = a, k \sim \rho_T)}{\mathbb{P}_{\tau \sim \mathcal{T}(x,\pi)}(A_0 = a | X_k = y, Z_k = z, k \sim \rho_T)} = \frac{\mathbb{P}_{\tau \sim \mathcal{T}(x,\pi)}(X_k = y, Z_k = z | k \sim \rho_T)}{\pi(a|x)}.$$

In addition, by the Markov property:

$$\mathbb{P}_{\tau \sim \mathcal{T}(x,\pi)}(A_0 = a | X_k = y, Z_k = z, k \sim \rho_T) = \mathbb{P}_{\tau \sim \mathcal{T}(x,\pi)}(A_0 = a | X_k = y, k \sim \rho_T),$$

$$= h_{\beta,T}(a|x,y).$$

Therefore:

$$\mathbb{P}_{\tau \sim \mathcal{T}(x,\pi)}(X_k = y, Z_k = z | A_0 = a, k \sim \rho_T) = \frac{h_{\beta,T}(a|x,y)\mathbb{P}_{\tau \sim \mathcal{T}(x,\pi)}(X_k = y, Z_k = z | k \sim \rho_T)}{\pi(a|x)}.$$

Thus, we can rewrite $Q^\pi(x,a)$ as:

$$Q^\pi(x,a) = r(x,a) + c_\gamma \sum_{y \in \mathcal{X}} \sum_{z \in \mathcal{X}} \frac{h_{\beta,T}(a|x,y)\mathbb{P}_{\tau \sim \mathcal{T}(x,\pi)}(X_k = y, Z_k = z | k \sim \rho_T)}{\pi(a|x)} r^\pi(z),$$

$$= r(x,a) + c_\gamma \sum_{k=1}^{T} \sum_{y \in \mathcal{X}} \sum_{z \in \mathcal{X}} \frac{h_{\beta,T}(a|x,y)\rho_T(k)\mathbb{P}_{\tau \sim \mathcal{T}(x,\pi)}(X_k = y)\mu_k(Z = z | X_k = y)}{\pi(a|x)} r^\pi(z),$$

$$= r(x,a) + \sum_{k=1}^{T-1} \gamma^k \sum_{y \in \mathcal{X}} \frac{h_{\beta,T}(a|x,y)\mathbb{P}_{\tau \sim \mathcal{T}(x,\pi)}(X_k = y)}{\pi(a|x)} r^\pi(y)$$

$$+ \gamma^T \sum_{y \in \mathcal{X}} \sum_{z \in \mathcal{X}} \frac{h_{\beta,T}(a|x,y)\mathbb{P}_{\tau \sim \mathcal{T}(x,\pi)}(X_k = y)}{\pi(a|x)} \tilde{d}^\pi(z|y) r^\pi(z),$$

$$= r(x,a) + \sum_{k=1}^{T-1} \gamma^k \sum_{y \in \mathcal{X}} \frac{h_{\beta,T}(a|x,y)\mathbb{P}_{\tau \sim \mathcal{T}(x,\pi)}(X_k = y)}{\pi(a|x)} r^\pi(y)$$

$$+ \gamma^T \sum_{y \in \mathcal{X}} \frac{h_{\beta,T}(a|x,y)\mathbb{P}_{\tau \sim \mathcal{T}(x,\pi)}(X_k = y)}{\pi(a|x)} V^\pi(y),$$

$$= r(x,a) + \mathbb{E}_{\tau \sim \mathcal{T}(x,\pi)} \left[ \sum_{k=1}^{T-1} \gamma^k \frac{h_{\beta,T}(a|x,X_k)}{\pi(a|x)} r^\pi(X_k) + \gamma^T \frac{h_{\beta,T}(a|x,X_T)}{\pi(a|x)} V^\pi(X_T) \right],$$

which concludes the proof. □

## D Algorithms

---
**Algorithm 1** State-conditional HCA
---
**Given:** Initial $\pi$, $h_\beta$, $V$, $\hat{r}$; horizon $T$
 1: **for** $k = 1, \ldots$ **do**
 2:     Sample $\tau = X_0, A_0, R_0, \ldots, R_T$ from $\pi$
 3:     **for** $i = 0, \ldots, T-1$ **do**                    ▷ Train hindsight distribution
 4:         **for** $j = i, \ldots, T$ **do**
 5:             Train $h_\beta(A_i | X_i, X_j)$ via cross-entropy
 6:         **end for**
 7:     **end for**
 8:     **for** $i = 0, \ldots, T-1$ **do**                  ▷ Train baseline and reward predictor
 9:         $Z = 0$
10:         **for** $j = i, \ldots, T-1$ **do**
11:             $Z \leftarrow Z + \gamma^{j-i} R_j$
12:         **end for**
13:         $Z \leftarrow Z + \gamma^{T-i} V(X_T)$
14:         Update $V(X_i)$ towards $Z$
15:         Update $\hat{r}$ towards $R_i$
16:     **end for**
17:     **for** $i = 0, \ldots, T-1$ **do**  ▷ Train policy of all actions with the hindsight-conditioned return
18:         **for** all actions $a$ **do**
19:             $Z_h = \pi(a|X_i, a)\hat{r}(X_i, a)$
20:             **for** $j = i+1, \ldots, T-1$ **do**
21:                 $Z_h \leftarrow Z_h + \gamma^{j-i} \frac{h_\beta(a|X_i, X_j)}{\pi(a|X_i)} R_j$
22:             **end for**
23:             $Z_{h,a} \leftarrow Z_h + \gamma^{T-i} \frac{h_\beta(a|X_i, X_T)}{\pi(a|X_i)} V(X_T)$
24:         **end for**
25:         Follow the gradient $\sum_a \nabla \pi(a|X_i) Z_{h,a}$
26:     **end for**
27: **end for**
---

---
**Algorithm 2** Return-conditional HCA
---
**Given:** Initial $\pi$, $h_z$, $V$
1: **for** $k = 1, \ldots$ **do**
2:     Sample $\tau = X_0, A_0, R_0, \ldots$ from $\pi$
3:     **for** $i = 0, 1, \ldots$ **do**
4:         Compose the return $Z(\tau_{i:\infty})$ starting from $X_i$
5:         Train $h_z(A_i|X_i, Z_i)$ via cross-entropy
6:         $Z_h \leftarrow \left(1 - \frac{\pi(A_i|X_i)}{h_z(A_i|X_i, Z(\tau_{i:\infty}))}\right) Z(\tau_{i:\infty})$
7:         Follow the gradient $\nabla \log \pi(A_i|X_i) Z_h$
8:     **end for**
9: **end for**
---

## E Experiment Details

The learning rate $\alpha$ for the baseline was chosen to be the best value from $[0.1, 0.2, 0.3, 0.4]$, while our model hyperparameters (the learning rate $\alpha_h$ for $h$, and the number of bins $n_b$ for the return version of HCA were selected informally to be $\alpha = 0.3, \alpha_b = 0.4, n_b = 3$ for the results in Fig. 4, and $n_b = 10$ elsewhere. Return HCA is sensitive to $n_b$, but all variants are robust to the choice of learning rate.

## F  Bootstrapping with state HCA

Consider the Delayed Effect task from Section 5, in which an action causes an outcome $T$ steps in the future, with everything in between being irrelevant. It is not immediately obvious why state HCA should be beneficial when one bootstraps with $n < T$. Indeed, if $h$ was perfect, the intermediate coefficient would be uninformative. However, we observe the opposite, precisely because $V$, $\pi$ and $h$ are being learned at the same time, but with different learning dynamics. In particular, in this case $h$ moves faster than $\pi$ (independently of the learning rate) as it is updated towards $1$ for any observed sample, while $\pi$ updates are modulated by the return. Now consider some interim $V(y) < 0$. The negative value implies that the policy at the initial state $x$ prefers the bad action $a$ over the good action $b$: $\pi(a|x) > \pi(b|x)$. But this in turn implies that $h(a|x,y)$ has been observed more frequently, and since $h$ is quicker to update: $h(a|x,y) > \pi(a|x)$. Now, take the policy gradient theorem (7) with $\pi$ as a baseline. The HCA return becomes $(h(a|x,y) - \pi(a|x))V(y) < 0$ and discourages the bad action. Similarly, $(h(b|x,y) - \pi(b|x))V(y) > 0$ and the good action is encouraged. We tested different learning rates, and initializations, and the effect persisted.