[Reviews · NeurIPS 2019]

Reviewer 1



Having read through the other reviews and the author response I will maintain my review of a 6. I really like the core idea of the paper and would be happy if it were accepted based on that alone. I appreciate the author's clarification of the experiments, and I now have a much clearer understanding of what was done. With that said I was a little disappointed that the answer to my question about bootstrapping was basically "a quirk of the learning dynamics". In general, the main reason I have not raised my score is that I found the significance of the experiments to be hard to judge and they don't necessarily clearly illustrate the merit of the approach. -This paper is fairly clear and well written all the way through -The main idea of this paper was to learn the impact of current actions on future trajectories as a means to improve credit assignment to actions and produce alternative advantage estimates which are suggested to be lower variance in many common situations. I found this idea to be very interesting and reasonable, and as far as I know fairly novel. -experiments are fairly clear and seem to do a reasonable job of testing certain properties of the proposed advantage estimators -Why show standard deviation in the error bars? wouldn't the standard error be more informative? -I found Figure 3 (right) a little confusing. If I'm reading it right it's showing the advantage estimate of taking the shortcut computed by different methods after training for 1000 episodes under a number of fixed policies. Is this correct? If so I think it could be written more clearly. Also in the text it says "Note that return conditioning is only able to do so when the preferred action is likely" but If I'm reading the graph correctly it shows that this is the case when the preferred action (shortcut) is *unlikely* instead, is this right? If so this should be fixed since that adds to the confusion. -In figure 4 (left) does HCA-state use bootstrapping or pure monte-carlo? If it uses Monte-carlo then it seems strange to compare with policy gradient using bootstrapping. This isn't showing anything about your proposed estimator but simply a limitation of bootstrapping under partial observability, which I presume the proposed method would suffer from too if it used bootstrapping. It is also unclear from the text whether HCA is using bootstrapping here or not. If it is I don't understand why it is able to perform better than policy gradient, could you offer an explanation if this is the case? -the baseline is just called "policy gradient", but since bootstrapping is discussed I assume this is an actor-critic algorithm? This could be clarified. -lack of explanation of how hyperparameters (such as learning rate) were selected in the experiments make the results difficult to interpret

Reviewer 2



The authors introduce a new reinforcement learning algorithm based on directly attempting to estimate attribution of credit. The authors achieve this by modeling the likelihood of a particular action a being taken at a state x given an outcome state y later in a trajectory. If the ratio of likelihoods between the P(a|y) and P(a|x). When the likelihood of a past action given a future state is much greater than the probability of that action being taken, then the state is highly predictive of a past action, and thus that past action was probably highly causal to the current state y. Using this ratio the authors are able to augment the typical SARSA setup to include this notion of credit assignment, and use it to improve training. The authors demonstrate the value of this new modeling in several experiments where they vary how much future states depend on past action, what is the delay, and noise confounding the setup. They are able to confirm the utility of the new modeling. The setup, motivation, and experimental details are very clear. The idea and formulation are original and highly central to many problems in reinforcement learning. The main drawback of the setup is a lack of a proper experiment or setup for how to achieve this result in a typical benchmark RL task (Atari, Mujoco, Gym-retro, etc..) where the benefits of this approach should also be visible. It is also difficult to know how to properly define and parametrize a function that predicts P(a|y), if the future state y contains some recurrent state from x which might memorise that a was taken (thus nullifying the value of this modeling). It is also important to know how to efficiently compute these functions -- in actor critic setups, it is sufficient to record the log probabilities of the taken actions and value function predictions and use those to perform updates (e.g. with clipping as done in PPO), however in this setup, given that it is necessary to compare the probability that at a past state a specific action was taken, what is the right way to model 'referring' back to a past state given future state information?

Reviewer 3



Originality: As far as I am aware this is a novel approach to estimating the value functions that has some potentially big advantages, as outlined in the paper (however, I am not familiar with niche literature in this vein). It takes a backward view of the learning problem and draws inspiration from importance sampling approaches to Monte Carlo estimation. This opens doors to new directions, which could prove useful. It seems like a hindsight distribution is something that humans naturally have access to, which is nice. I should note that the direction taken is not incremental, but more of a blue ocean idea---a true alternative. This is what makes this a “high risk, high reward” paper from my perspective. The related work section is adequate, although connections/parallels to other works could be clearer. E.g., it is unclear to me what the connection to hindsight experience replay is (besides the use of the word “hindsight”). Quality: I went through the main proofs/derivations (with the exception of the derivation of the beta hindsight distribution) and things appear technically sound. There are no extravagant/unsupported claims made, although the discussion in Subsection 4.3 failed to persuade me that learning the hindsight distribution would be easy in non-toy tasks (esp. as it is policy dependent). I would say the present work is a complete package, although it would benefit greatly from at least one non-toy experiment. The authors frame this as a novel idea/framework and do not make empirical claims, so I am OK with the lack of non-toy results. Clarity: This paper is well organized and a pleasure to read. My only comments/suggestions/questions are as follows: Line 101: Why is h_k a higher entropy distribution in general? Line 192: extra space Section 5: Although these are toy experiments, I think not enough information is given to reproduce (missing, e.g., learning rate used, and hyperparameters tested) Why is Figure 4(center) cut off at 200 episodes? Does MC PG overtake both HCA curves? Significance: I discussed significance in the Contributions+Originality sections. To summarize: IMO this is an interesting, novel direction that is worth pursuing. As the idea was not validated in any non-toy example, I would rate the immediate significance as low, but believe this could have very high significance if it successfully scales / handles non-toy tasks. But even if it fails to scale, I would like this to be a part of the literature, as it provides a new perspective/form for the value function and may inspire related ideas. *** Additional comments after author response *** - I found the author response helpful on minor points, although it does not address the lack of a proper experiment or persuade me that the hindsight distributions can be learned. I think any proper task, even a simple Gridworld, would make this a much better paper. For this reason, I am maintaining my original score.

[Author Response · NeurIPS 2019]

We thank all reviewers for their time and great feedback. We'll incorporate various suggestions and clarifications in the
revision. Here, we first address the shared points, then individual comments.

**Hyperparameters (R1 and R3).** The learning rate $\alpha$ for the baseline was chosen to be the best value from
$[0.1, 0.2, 0.3, 0.4]$, while our model hyperparameters (the learning rate $\alpha_h$ for $h$, and the number of bins $n_b$ for
the return version of HCA) were selected informally to be $\alpha = 0.3, \alpha_b = 0.4, n_b = 3$ for the results in Fig. 4, and
$n_b = 10$ elsewhere. Return HCA is sensitive to $n_b$, but all variants are quite robust to the choice of learning rate. We'll
report all of this.

**Learning $h$ at scale (R2 and R3).** We are working on this. The simplest architecture is a standard A3C agent with
an extra layer that takes the embeddings $x, x'$ (e.g. outputs of the conv layers) of two observations as an input, and
outputs a distribution over actions $h(\cdot|x, x')$. It can be trained with the cross-entropy loss on all $x, x', a$ samples on
observed trajectories of complete episodes. (For very long episodes, one may need to subsample). The return version is
similar, but simply takes the return value as the second input. Alternatively, one could use ideas from noise contrastive
estimation, and parametrize the ratio $h/\pi$ directly, similarly e.g. to the recent CPC algorithm. The positive examples
would be actions on observed trajectories, and the negative examples – independent samples from the current $\pi$. To use
$h$, like in regular A3C, one needs to remember the trajectory up to unroll length. In the state version, the return cannot
be recursively composed anymore, and so the complexity of the update becomes quadratic in the length of the trajectory
(return version remains linear). Indeed, $h$ is policy dependent, but our intuition is that it can be meaningful / helpful
even without being exactly correct for a particular policy. See e.g. the response to R1.

**Reviewer 1** Why show standard deviation in the error bars? wouldn't the standard error be more informative?

We care about how robust the learning performance is, the std of 100 independent runs captures the deviation in
performance run-to-run, while the sem would measure the confidence in the mean (but not how likely it is to get it).

I found Figure 3 (right) a little confusing.

Thanks for spotting this, you are entirely correct. We will clarify and fix this.

In figure 4 (left) does HCA-state use bootstrapping? If it is I don't understand why it is able to perform better.

Thanks for the great comment! All variants use the same $n$ and bootstrap, so it really is about using HCA. We spent
a considerable amount of time working out exactly what it is that makes the state version learn here, since (as you
rightly point out), if everything was learned perfectly the intermediate ratios would be 1, and no learning would happen
at all. The key is that we are learning $h$, $\pi$ and $V$ at the same time, but their learning dynamics are different. In
particular $h$ moves quicker than $\pi$ (regardless of learning rate) as it is updated towards 1 for any observed sample.
Now consider some interim $V(y) < 0$. That means the policy at the initial state $x$ prefers the bad action $a$ over good
action $b$: $\pi(a|x) > \pi(b|x)$. But this also means that $h(a|x, y)$ has been observed more frequently, and because the cross
entropy loss is more aggressive: $h(a|x, y) > \pi(a|x)$. Therefore the HCA return $= (h(a|x, y)/\pi(a|x) - 1)V(y) < 0$
and *discourages* the bad action. Similarly, $(h(b|x, y)/\pi(b|x) - 1)V(y) > 0$ and the good action is encouraged. We
tested different learning rates, and initializations, and the effect persisted. We'll add this discussion to the paper.

I assume this is an actor-critic algorithm?

Indeed, all baselines implement $n$-step advantage actor critic, with $n = \infty$ for Monte Carlo.

**Reviewer 2** How to properly define and parametrize a function that predicts P(a|y), if the future state y remembers a

This is a great point, and something we are thinking about. Note that the return version doesn't suffer from this. We
could consider other forms of future conditioning that are richer than the return and remain informative when past
actions affect the representation.

**Reviewer 3** what is the connection to hindsight experience replay (besides the use of the word "hindsight").

It really is mostly the word :) The idea behind HER is to use a trajectory $\tau$ to train not only with the goal pursued by the
policy that generated $\tau$, but also other (randomly sampled) goals (counterfactual goals), whereas we are concerned with
efficient credit assignment for the same goal (counterfactual actions).

Why is $h_k$ a higher entropy distribution in general?

Good catch! That's a typo and should say *lower* (or equal) entropy. This is because we never add uncertainty by
conditioning on an additional random variable, so the result is a sharper distribution.

Why is Figure 4(center) cut off at 200 episodes? Does MC PG overtake both HCA curves?

This is simply an oversight. All variants reach the same asymptotic performance. We'll make the number of episodes
consistent in the final version.

[Meta-Review · NeurIPS 2019]

The reviewers found the submission interesting, novel and potentially of significant impact, although this was tempered somewhat by only limited empirical support.